# Identification of Novel Fusion Genes in Bone and Soft Tissue Sarcoma and Their Implication in the Generation of a Mouse Model

**DOI:** 10.3390/cancers12092345

**Published:** 2020-08-19

**Authors:** Yasuyo Teramura, Miwa Tanaka, Yukari Yamazaki, Kyoko Yamashita, Yutaka Takazawa, Keisuke Ae, Seiichi Matsumoto, Takayuki Nakayama, Takao Kaneko, Yoshiro Musha, Takuro Nakamura

**Affiliations:** 1Division of Carcinogenesis, The Cancer Institute, Japanese Foundation for Cancer Research, 3-8-31 Ariake, Koto-ku, Tokyo 135-8550, Japan; yasuyo.teramura@jfcr.or.jp (Y.T.); miwa.tanaka@jfcr.or.jp (M.T.); yukari.yamazaki@jfcr.or.jp (Y.Y.); 2Division of Pathology, The Cancer Institute, Japanese Foundation for Cancer Research, 3-8-31 Ariake, Koto-ku, Tokyo 135-8550, Japan; kyoko.yamashita@jfcr.or.jp (K.Y.); yutaka.takazawa@jfcr.or.jp (Y.T.); 3Department of Pathology, Toranomon Hospital, Tokyo 105-8470, Japan; 4Division of Orthopedic Oncology, The Cancer Institute Hospital, Japanese Foundation for Cancer Research, 3-8-31 Ariake, Koto-ku, Tokyo 135-8550, Japan; keisuke.ae@jfcr.or.jp (K.A.); smatsumoto@jfcr.or.jp (S.M.); 5Department of Orthopedic Surgery (Ohashi), School of Medicine, Toho University, Tokyo 143-8540, Japan; takayuki.nakayama@med.toho-u.ac.jp (T.N.); takao-knee@oha.toho-u.ac.jp (T.K.); musha@oha.toho-u.ac.jp (Y.M.)

**Keywords:** fusion gene, bone and soft tissue sarcomas, target RNA sequencing, mouse model, NTRK, inhibitor

## Abstract

Fusion genes induced by chromosomal aberrations are common mutations causally associated with bone and soft tissue sarcomas (BSTS). These fusions are usually disease type-specific, and identification of the fusion genes greatly helps in making precise diagnoses and determining therapeutic directions. However, there are limitations in detecting unknown fusion genes or rare fusion variants when using standard fusion gene detection techniques, such as reverse transcription-polymerase chain reaction (RT-PCR) and fluorescence in situ hybridization (FISH). In the present study, we have identified 19 novel fusion genes using target RNA sequencing (RNA-seq) in 55 cases of round or spindle cell sarcomas in which no fusion genes were detected by RT-PCR. Subsequent analysis using Sanger sequencing confirmed that seven out of 19 novel fusion genes would produce functional fusion proteins. Seven fusion genes detected in this study affect signal transduction and are ideal targets of small molecule inhibitors. *YWHAE-NTRK3* expression in mouse embryonic mesenchymal cells (eMCs) induced spindle cell sarcoma, and the tumor was sensitive to the TRK inhibitor LOXO-101 both in vitro and in vivo. The combination of target RNA-seq and generation of an ex vivo mouse model expressing novel fusions provides important information both for sarcoma biology and the appropriate diagnosis of BSTS.

## 1. Introduction

Bone and soft tissue sarcomas (BSTS) are rare malignant neoplasms consisting of more than 50 independent diseases [1]. Approximately 7000 new BSTS patients are observed every year in Japan [2,3]. Pathological diagnosis of BSTS is sometimes challenging even for expert pathologists when the case is in a very rare disease category. Fusion genes caused by chromosomal aberrations are observed in 30% of BSTS, and detection of specific fusion genes greatly helps improve diagnosis [1,4,5,6]. Moreover, these fusion genes are targets or potential targets of effective drug therapies [5]. Indeed, the identification of novel fusion genes such as *CIC-DUX4* and *BCOR-CCNB3* has resulted in establishment of new disease entities, which has provided new insights into sarcoma biology and disease classification [1,7,8].

Advances in next-generation sequencing (NGS) technologies have enabled efficient identification of unknown fusion genes in BSTS [4,6]. Among NGS technologies, target RNA sequencing (RNA-seq) utilizes sequences of known fusion gene partners as probes and provides highly sensitive detection of both novel and known fusion genes [6,9]. Reverse transcription-polymerase chain reaction (RT-PCR), with or without fluorescence in situ hybridization (FISH), is useful for detecting classical fusion genes in BSTS, such as synovial sarcoma and myxoid liposarcoma, in which limited variations of fusion genes exist [10,11]. However, these methods are limited in detecting fusion genes by the fact that one of the partners remains unknown. Our laboratory has been engaging in the molecular diagnosis of BSTS since 2003 at the Cancer Research Hospital of Japanese Foundation for Cancer Research. Despite efforts to identify most of the known fusion genes, there remain BSTS cases in which fusion genes have not been detected by these conventional methods. By using target RNA-seq, we have identified novel and known fusion genes in BSTS that were previously related to unclassified sarcomas or were diagnosed incorrectly.

Once certain novel fusion genes have been identified, it is ideal to generate a mouse model that bears the fusion gene expressed in the appropriate cell lineage if similar morphologies to the original human sarcoma are exhibited. We have previously generated unique mouse model systems for fusion gene-associated BSTS by retrovirally introducing genes into mouse embryonic mesenchymal cells (eMCs). These BSTS include Ewing sarcoma, alveolar soft part sarcoma, synovial sarcoma and CIC-DUX4 sarcoma [12,13,14,15], and our mouse models recapitulate both morphological and genetic profiles of the human counterparts. In this study, we have generated a novel mouse model for novel *NTRK3* fusion gene-associated sarcomas, and tested the therapeutic efficiency of the NTRK inhibitor, LOXO-101. The significant growth suppressive effect of LOXO-101 in *YWHAE-NTRK3*-expressing sarcomas in mice indicates that modeling sarcomas is a promising tool to help understand the biological significance of novel fusion genes and to evaluate novel drugs for BSTS.

## 2. Results

### 2.1. Disease Characteristics

A total of 55 cases of BSTS were subjected to target RNA-seq analysis (Table 1 and Appendix A). These cases consisted of 30 spindle cell sarcomas and 25 round cell sarcomas, ostensibly without differentiation toward known mesenchymal lineages (Figure 1). In some cases, the original diagnosis was a solitary fibrous tumor, synovial sarcoma or Ewing sarcoma, with which known fusion genes are associated, although these diagnoses were not confirmed with RT-PCR (Table 1). Of the total cases subjected to RNA-seq analysis, 24 were female and 31 were male, with a wide age range at diagnosis (13–89 years, median age: 48 years). BSTS of the examined cases were widely distributed anatomically, with six cases in bone and the others in soft tissues. Thirteen of 25 round cell sarcoma cases, compared to 20 of 30 spindle cell sarcomas, had developed in the extremities. Eight round cell sarcoma and 10 spindle cell sarcoma cases had developed in the trunk.

### 2.2. Fusion Gene Profiles of 29 BSTS Cases

Fusion genes were detected in 29 of the 55 BSTS cases by target RNA-seq, with 19 of those being spindle cell sarcomas and 10 being round cell sarcomas. These 29 cases included 14 females and 15 males, with a median age of 51 years. There were no significant differences in sex and age between fusion gene-positive and -negative cases. Notably, eight of 12 spindle cell sarcoma cases, for which the original diagnosis was a solitary fibrous tumor, were positive for fusion genes. A total of 47 fusion genes were detected in 29 cases (Figure 1, Table 2 and Appendix A). Multiple gene fusions within a single case were observed, with two fusions in eight cases, three fusions in two cases and seven fusions in one case. All 47 fusion genes were subjected to validation by RT-PCR followed by Sanger sequencing, and 35 were also detected and confirmed by both of these methods. Twenty of 35 fusion genes were potentially generated by interchromosomal translocations, and 15 were potentially made by intrachromosomal aberrations such as inversions and/or deletions. These 35 fusion genes included 19 novel fusion genes, though these fusions could be generated by other mechanisms such as aberrant splicing [16]. To confirm whether the fusion was novel or known, the candidate genes were subjected to database searching using two independent databases for fusion genes: Mitelman Database Chromosome Aberrations and Gene fusions in Cancer (https://mitelmandatabase.isb-cgc.org) and Tumor Fusion Gene Data Portal (https://tumorfusions.org).

Sequence analysis showed that seven of 19 novel fusion genes potentially produce functional fusion proteins resulting from in-frame fusions (Table 2). There were also 16 known fusion genes that were not detected during our previous RT-PCR examination due to difficult pathological and/or clinical diagnostic difficulties in predicting the correct fusion genes, difficult sequences for amplification in RT-PCR or rare exon combinations of two genes (Appendix A). Target RNA-seq thus identified not only novel fusion genes, but also revealed clinically and pathologically unexpected known fusion genes. These unexpected fusion genes include three cases of *NAB2-STAT6*, two cases of *EWSR1-FLI1* and a single case of each of *EWSR1-ATF1, EWSR1-NR4A3, EWSR1-CREB1, CIC-DUX4, SS18-SSX1* and *GAB1-ABL1*.

### 2.3. Deduced Structure of Novel Fusion Proteins

Both target RNA-seq and Sanger sequencing analyses clarified fused exons of both genes (Figure 2 and Figure 3).

#### 2.3.1. AHRR-NCOA3 (Case #43)

*AHRR-NCOA3* was identified in a 60-year-old female case of spindle cell sarcoma, originally diagnosed as a solitary fibrous tumor. Recurrent fusions between *AHRR* and *NCOA2,* an *NCOA3* homolog, by t (5;8) (p15;q13), have been reported in angiofibroma of soft tissue [17], which was also identified in case #45, for which the original diagnosis was hemangiopericytoma. Similarly to the *AHRR-NCOA2* fusion, the sequence analysis showed that *AHRR-NCOA3* encodes a protein possessing bHLH and PAS domains of AHRR, and two transactivation domains for the steroid hormone receptor and CREB-interacting domain of NCOA3 (Figure 2A), suggesting close functional similarity to transcriptional activators between the two fusion proteins. Spindle cell proliferation and thin wall capillary vessels with inflammatory infiltrate is consistent with what is found in angiofibroma of soft tissue (Figure 2A) [1].

#### 2.3.2. PAK2-RAF1 (Case #2)

In a previous study, RAF1 fusions were reported in soft tissue sarcoma, in which the fusion counterparts, *PDZRN3, SLMAP* and *TMF1*, were identified in three of eight cases [18]. *RAF1* and its four fusion partners, including *PAK2,* are located within chromosome 3, indicating that *RAF1* fusions result from intrachromosomal rearrangements. The protein kinase domain of RAF1 is preserved in PAK2-RAF1 (Figure 2B), as was indicated in the previous study [18], suggesting that the fusion protein may function in the receptor/MAPK signaling. It is likely that replacement of the RAS-binding domain of RAF1 by the p21-Rho binding domain of PAK2 significantly modulates the RAS/RAF signaling and may induce upregulation of the downstream molecules of the pathway. The morphology of the tumor in case #2 is that of small round cell sarcoma; however, its rather low cellularity does not strongly suggest an aggressive nature (Figure 2B).

#### 2.3.3. EPC1-KDM2B (Case #55)

EPC1 (enhancer of polycomb 1) is a component of the Nu4A histone acetyltransferase complex, which promotes E2F1 transcriptional activity [19,20]. *EPC1* is found fused to *PHF1* in endometrial stromal sarcomas and ossifying fibromyxoid tumors [21,22], and the present study also identified the *EPC1-PHF1* fusion in the round cell sarcoma case #28. KDM2B is a member of the polycomb repressive complex (PRC1.1) and interacts with SS18-SSX1 [23]. Deregulation of PRC1.1 was reported in various human malignancies, such as gynecologic cancer and myelodysplastic syndrome [24]. As shown in Figure 2C, conservation of the EPC1 protein interaction domain of EPC1 as a NuA4 member and the KDM2B catalytic domain suggests that fusion affects the functions of both the Nu4A and PRC1.1 complexes. No ossification and scar distribution processes of spindle cells were distinct from those found in ossifying fibromyxoid tumors (Figure 2C).

#### 2.3.4. TAOK1-CRYBA1 (Case #14)

*TAOK1* is upregulated in a group of breast cancer and found fused in breast cancer cell lines [25,26]. *CRYBA1* encodes crystallin bA1, a structural protein that is required for the lens of the eyes, being involved in lysosomal functions, and is expressed in lens and retinal pigmented epithelium [27,28,29]. In the present study, most of the functional TAOK1 domains were lost and a large part of CRYBA1 was found to remain in the fusion protein (Figure 3A), suggesting that CRYBA1 may contribute to the function of the fusion protein. The function and involvement of *TAOK1-CRYBA1* in cancer has not been reported, and its functional significance should be determined. Aggressive growth of undifferentiated tumor cells with bizarre nuclei did not provide information on tumor classification (Figure 3A).

#### 2.3.5. UST-DUSP22 (Case #50)

The deduced amino acid sequence predicted that UST-DUSP22 consists of largely of DUSP22 and a little of UST (Figure 3B). *DUSP22* is involved in chromosomal translocation in cases of ALK-negative anaplastic malignant lymphoma [30]. *UST* encodes uronyl 2-sulfotransferase [31]; however, the functional domain is not preserved in UST-DUSP22, indicating a small contribution of UST in the fusion protein function. Since the tumor suppressive role of *DUSP22* has been proposed [32], it is intriguing to consider whether *UST-DUSP22* possesses a dominant negative effect against wild-type *DUSP22*. It is currently unknown whether spindle cell proliferation in this case is related to the findings observed in ALK-positive tumors such as inflammatory myofibroblastic tumor (Figure 3B).

#### 2.3.6. YWHAE-NTRK3 (Case #32) and PPFIBP1-NTRK3 (Case #42)

The present study identified four cases of *NTRK* fusions consisting of *TPR-NTRK1* in two cases, *YWHAE-NTRK3* and *PPFIBP1-NTRK3*. *YWHAE* encodes a protein of the 14-3-3 family, and has been found fused to *FAM22*, or *NUTM2A/B/E,* in endometrial stromal sarcoma [33,34] and clear cell sarcoma of the kidney [35]. *PPFIBP1* has been found fused to *ALK* in pulmonary inflammatory myofibroblastic tumors [36]. In both NTRK3 fusions herein, the protein kinase domain was preserved in the fusion proteins (Figure 4A,B), and the transforming capacity of *PPFIBP1-ALK* suggests that these NTRK3 fusions upregulate signaling downstream of NTRK3. Morphological analysis of the cases with the *YWHAE-NTRK3* or *PPFIBP1-NTRK3* fusions showed proliferation of short spindle tumor cells with occasional storiform pattern (Figure 4A,B). Diffuse expression patterns of tumor cells with the anti-CD34 and anti-pan-TRK antibodies were observed in the *YWHAE-NTRK3* case by immunostaining (Figure 4C,D). These histological and immunohistochemical findings are consistent with those previously observed for NTRK-rearranged spindle cell neoplasm [1,19].

### 2.4. Generation of an Animal Model for NTRK3 Fusion-Associated Sarcomas

To understand the tumorigenic functions of BSTS-associated fusion genes and to test new therapeutic tools, it is important to establish animal models for introduction of fusion genes into appropriate cell types; however, it is often difficult to develop sarcomas in mice that are phenotypically representative of human diseases. Using retrovirus-mediated gene transfer and mouse eMCs, we have previously succeeded in creating ex vivo mouse models for human sarcomas [12,13,14,15]. Using this technology, we tried to generate mouse models expressing kinase-related genes identified in this study, including *YWHAE-NTRK3, PPFIBP1-NTRK3, PAK2-RAF1*, *GAB1-ABL1* and wild-type *ERAS*. The full-length cDNAs of each gene were cloned into the pMY retrovirus vector and introduced into E18.5 eMCs of limb and trunk soft tissue prepared from Balb/c mice. Retrovirally transduced eMCs were immediately transplanted into the subcutaneous parts of Balb/c nude mice. Among five experimental lines, five of thirteen (38%) recipient mice bearing *YWHAE-NTRK3*-expressing eMCs developed subcutaneous tumors (Figure 5A,B). Morphological analysis of the mouse tumor with the YWHAE-NTRK3 fusion showed proliferation of short spindle tumor cells with fibromyxoid stroma (Figure 5C). The tumor was rich in blood vessels; however, eosinophilic hyaline changes were not observed, unlike in human sarcomas. The tumor showed strong positive staining for NTRK by immunohistochemistry using an anti-pan-TRK antibody (Figure 5D). The tumor exhibited invasive growth into the surrounding tissue and could be serially transplanted to nude mice, yet no distant metastasis was observed.

### 2.5. The Effect of the NTRK Inhibitor on the YWHAE-NTRK3 Sarcoma Model

Appropriate animal models for human cancer provide useful platforms to test the effects of therapeutic drugs as preclinical models. The *YWHAE-NTRK3*-induced mouse sarcoma model therefore provides a good system to evaluate molecular target drugs for *NTRK* fusion genes. *YWHAE-NTRK3*-expresssing mouse sarcoma cells were treated in vitro with the pan-NTRK inhibitor, LOXO-101 [37]. The treatment inhibited cell proliferation significantly, with an IC_50_ of 42.1 nM. (Figure 6A). LOXO-101 treatment suppressed DNA synthesis and induced apoptosis of sarcoma cells (Figure 6B,C), indicating that sarcoma cells are highly dependent on upregulated NTRK signaling. Immunoblotting showed suppression of phosphorylation in the NTRK fusion protein as well as ERK, with little effect on AKT phosphorylation (Figure 6D), suggesting that the MEK/ERK signaling is a major downstream component of *YWHAE-NTRK3*-induced phosphorylation. The growth inhibitory effect of LOXO-101 on *YWHAE-NTRK3*-expressing sarcomas was further examined in vivo. *YWHAE-NTRK3*-expressing sarcoma cells were transplanted into nude mice and 100 mg/kg of LOXO-101 was orally administered for a week, starting 12 days after transplantation. Significant and rapid regression of tumor growth was observed after LOXO-101 treatment (Figure 6E), clearly indicating that the *NTRK3* fusion-associated sarcoma is sensitive to this specific inhibitor.

## 3. Discussion

Target RNA-seq is a highly efficient and sensitive method for detecting cancer-associated fusion genes. This method identified the *EWSR1-FLI1* fusion in an Ewing sarcoma case (case #8), in which it was difficult to detect the fusion by RT-PCR, probably due to high GC content of the target region. Target RNA-seq is also useful to detect fusions in solitary fibrous tumors that show multiple variants of fusion transcripts, such as *NAB2-STAT6* [38,39]. Certain tumor types, such as NTRK-rearranged spindle cell neoplasm and nodular fasciitis, are known to exhibit fusions with multiple fusion partners for invariable partner genes such as *NTRK1/2/3* and *USP6* [40,41]. In this study. we identified two known *NTRK1* fusions, two novel *NTRK3* fusions and the recently discovered *USP6* fusion associated with malignant nodular fasciitis [42]. Detection of known fusion genes such as *EWSR1-ATF1* (case 1), *GAB1-ABL1* (case 31) and *EWSR1-CREB1* provided useful information to re-consider original diagnosis (Appendix A and Appendix A). Thus, rapid detection of these types of fusions is a significant merit of targeted RNA-seq.

It has been reported that most of the fusion products of BSTS function in transcriptional modulation [43]; however, the identification of *NTRK* fusions, as well as *PAK2-RAF1, GAB1-ABL1* and *MSN-ERAS* in this study, indicates that more than a few fusion genes of BSTS are involved in inducing the upregulation of signal transduction. To explore the functional significance of these kinase-associated fusions, we have tried to generate mouse models by introducing fusion genes into mouse embryonic mesenchymal cells. Using this method, we have successfully produced four mouse models of human BSTS pertaining to Ewing sarcoma, alveolar soft part sarcoma, CIC-DUX4 sarcoma and synovial sarcoma [12,13,14,15]. All models exhibited representative morphologies and gene expression profiles for human tumors. In this study, the expression of *YWHAE-NTRK3* induced sarcomas at 40% penetrance, whereas that of four other fusions, namely *PPFIBP1-NTRK3, PAK2-RAF1*, *GAB1-ABL1* and wild-type *ERAS*, did not. Relatively low efficiency in tumor induction may be due to rare cell-of-origin populations in mouse embryonic mesenchymal cells. Alternatively, the oncogenic activity of fusion genes is variable and additional mutations may be required for certain fusion genes. We previously found that the cooperative upregulation of *miR-214* accelerated *SS18-SSX1*-induced development of synovial sarcoma [15]. Given the importance of such cooperative events in human dedifferentiated liposarcoma [44], the combination of fusion gene expression and cooperative gene mutations should be tested in future, which would provide further insights for understanding the development mechanisms of BSTS and fusion gene functions.

Our new model for NTRK-rearranged spindle cell neoplasm showed fibrosarcoma-like morphologies enriched with capillary blood vessels, which is consistent with human disease. *YWHAE-NTRK3*-expressing sarcoma cells showed rapid growth in vitro and were serially transplantable to nude mice. The YWHAE-NTRK3 fusion protein was highly phosphorylated, resulting in the upregulation of downstream signals such as ERK. *YWHAE-NTRK3*-expressing mouse sarcoma is highly dependent on NTRK3 signaling, which was evident from the effective growth suppression of the NTRK inhibitor LOXO-101. Moreover, in vivo tumor growth was almost completely abolished by LOXO-101. The growth suppressive effect of LOXO-101 in vivo has also been reported in a PDX model of *NTRK1/3* fusions [37]. Confirmation of this effect using our genetically engineered tumor cell-derived allograft model can contribute to molecular targeted drug development.

This study also identified novel fusion genes, the functions of which remain unknown. One such fusion is *TAOK1-CRYBA1*, which has the structural protein crystallin-b expressed in retinal astrocytes and required for eye lens function [27,28]. *TAOK1-CRYBA1* was identified based on the rationale that *TAOK1* upregulation is related to the oncogenic potential in malignancies such as breast cancer [26,27]. Therefore, the current method may reveal unexpected novel functions of fusion partner genes in cancer development and progression. In addition, sequence analysis of several fusion genes failed to show functional products due to out-of-frame fusions or loss of functional domains. These malfunctional fusions were found in *YTHDF2-GMEB1, BTBD9-HMGA2, BTBD9-PVT1, PTPRR-KSR2, TBX5-PTPRR, DDX26B-IRF4, NCOA2-SERPINA7, CNNM2-EGR2, RPSAP52-TGFBR3, FRYL-FIP1L1* and *CRTC1-DOHH* (Appendix A). It is currently unknown whether these fusions affect tumorigenesis or if they are passenger mutations, and further study is needed to clarify their natures. Cancer-related chromosomal instability induces chromothripsis, resulting in random gene fusions [45]. The *HMGA2* locus is affected in mesenchymal tumors, including adipogenic tumors and uterine leiomyoma [46,47], suggesting that the locus may be a fragile target for dynamic chromosomal rearrangement, especially in case #52.

## 4. Materials and Methods

### 4.1. Patients and Clinical Samples

A total of 55 BSTS patients either operated on or biopsied at the Japanese Foundation for Cancer Research hospital were used for this study. Cases were selected from the archive of the Sarcoma Center at Japanese Foundation for Cancer Research, as no fusion gene was detected by the RT-PCR method (Table 1 and Appendix A). Two pathologists (K.Y. and Y.T.) reviewed the histology of all samples to confirm the diagnosis. The study was conducted in accordance with ethical guidelines and approved by the Institutional Review Board at the Japanese Foundation for Cancer Research under license 2013-1155.

### 4.2. Targeted RNA Sequencing and Validation of Fusion Gene Expression by RT-PCR

The targeted sequencing library was prepared using a TruSight RNA Fusion Panel (Illumina, San Diego, CA, USA) according to the manufacturer’s protocol. Briefly, the total RNA extracted from the fresh frozen tissue samples was reverse-transcribed, and complementary DNA was hybridized with oligonucleotide probes for the target genes. RNA sequencing with 2 × 75 bp paired-end reads was performed on MiSeq (Illumina). Fusion genes were analyzed using RNA-Seq Alignment (BaseSpace Workflow version 1.1.0, Illumina). Fusion calls were made if there were at least three unique reads that meet the quality metrics as follows: split reads plus paired reads ≥3; fusion contig alignment length ≥16 bp; break-end homology ≤10 bp; coverage after fusion ≥100 bp. Each fusion gene was confirmed by RT-PCR using the same RNA sample, followed by Sanger sequencing using specific primers, as shown in Appendix A.

### 4.3. Histopathology and Immunohistochemistry

Formaldehyde-fixed, paraffin-embedded tumor tissues were stained with hematoxylin and eosin (H&E) using standard techniques. Immunohistochemical staining was performed with a rabbit monoclonal anti-pan-TRK antibody (ab181560; Abcam, Cambridge, UK) and a mouse monoclonal anti-CD34 antibody (clone NU-4A1; Nichirei, Tokyo, Japan). Microscopic images were obtained and analyzed by an Olympus BX53 microscope (Olympus, Tokyo, Japan). Images were acquired with a DP73 CCD camera (Olympus) using cellSens software (Olympus) and were processed using Adobe Photoshop version 21.0.1.

### 4.4. Generation of the Model Mouse Expressing Novel Fusion Genes

Mouse models were generated according to a method described previously [12,13,14,15]. Briefly, the *YWHAE-NTRK3, PPFIBP1-NTRK3, PAK2-RAF1, GAB1-ABL1* and *ERAS* cDNAs were synthesized using total RNA extracted from the patients’ samples. These full-length cDNAs were FLAG-tagged and introduced into the pMYs-FLAG-IRES-GFP retroviral vector. Embryonic mesenchymal cells (eMCs) were prepared aseptically from limb or trunk soft tissues of an 18.5 dpc Balb/c mouse embryo (Clea Japan, Tokyo, Japan). eMCs were immediately subjected to retroviral spin infection over two days. Transduced eMC cells (1 × 10^6^) were mixed with Matrigel (BD Biosciences, Franklin Lakes, NJ, USA) and transplanted into the subcutaneous regions of Balb/c nude mice (Clea, Tokyo, Japan). Recipients were carefully observed, and subcutaneous tumors were removed when they reached 10 mm in diameter. Cell lines were established from subcutaneous primary tumors and were maintained in Iscove’s Modified Dulbecco’s Medium supplemented with 10% fetal bovine serum.

### 4.5. Pharmacological Experiments

*YWHAE-NTRK3*-expressing murine sarcoma cells were treated with 50 nM of LOXO-101 (Selleck Chemicals, Houston, TX, USA) in vitro. Cells were seeded into 96-well plates at a concentration of 5 × 10^3^ cells per well and were treated with drugs for 72 h. XTT assays were then performed, and the half-maximal concentration (IC_50_) was calculated. Fluorescence-activated cell sorting (FACS) analysis was performed for 24 h after treatment with drugs to detect Annexin-V- or EdU-positive fractions. For in vivo experiments, 1 × 10^6^ cells were transplanted subcutaneously into Balb/c nude mice, and the mice were treated with 100 mg/kg of LOXO-101 via oral administration daily, for seven days.

### 4.6. Western Blotting

Protein samples were prepared in radioimmuno@recipitation (RIPA) buffer containing a complete protease inhibitor cocktail. The antibodies used were FLAG (F3165; Sigma-Aldrich, St. Louis, MO, USA), phospho-TRK (ab197071; Abcam), phospho-AKT (#4060; Cell Signaling Technology, Danvers, MA, USA), pan-AKT (#4691; Cell Signaling Technology), phospho-p44/42 MAPK (#9101; Cell Signaling Technology), p44/42 MAPK (#9102; Cell Signaling Technology) and α-tubulin (T5168; Sigma-Aldrich). The whole western blot images have been shown in Appendix A. 

### 4.7. Flow Cytometry

Apoptosis of tumor cells was analyzed using an Annexin V-633 Apoptosis Detection Kit (Nakalai, Kyoto, Japan). Cell proliferation analysis was performed using The Click-iT EdU Cell Proliferation Kit (Thermo Fisher Scientific, Waltham, MA, USA). Subsequently, 10 mM of EdU was added 1 h before cell collection. Flow cytometry was carried out using the BD FACSLyric system (BD Bioscience).

### 4.8. Statistics

All in vitro experiments were performed at least in triplicate. The numbers of mice used per experiment are indicated in the text or figure legends. Values are expressed as mean ± SEM, and statistical significance was determined using a two-tailed Student’s *t*-test.

## 5. Conclusions

Target RNA-seq identifies novel fusion genes effectively in BSTS diagnosed as tumors of unknown categories or misdiagnosed as different types, which resulted in the correction of diagnoses and will contribute to therapeutic directions. Modeling fusion gene-associated sarcoma using the ex vivo gene transfer technique succeeded in generating the *NTRK3* fusion sarcoma model and provided useful information for molecular targeted therapy as a preclinical model, though improvement of the method is needed for more efficient tumor induction. Selection of a more appropriate cell-of-origin and addition of cooperative genetic events will not only greatly improve this method, but also provide insights into the cell-of-origin of BSTS.

## Figures and Tables

**Figure 1 cancers-12-02345-f001:**
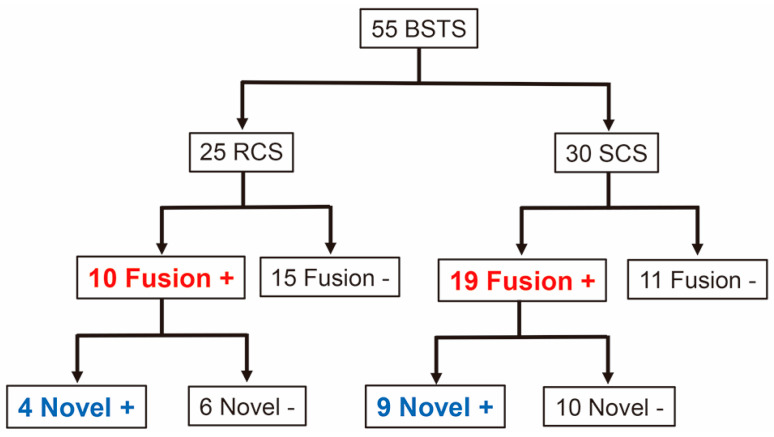
Summary of the 55 BSTS cases subjected to target RNA-seq analysis. Distribution of novel fusions are indicated as +. RCS: round cell sarcomas; SCS: spindle cell sarcomas.

**Figure 2 cancers-12-02345-f002:**
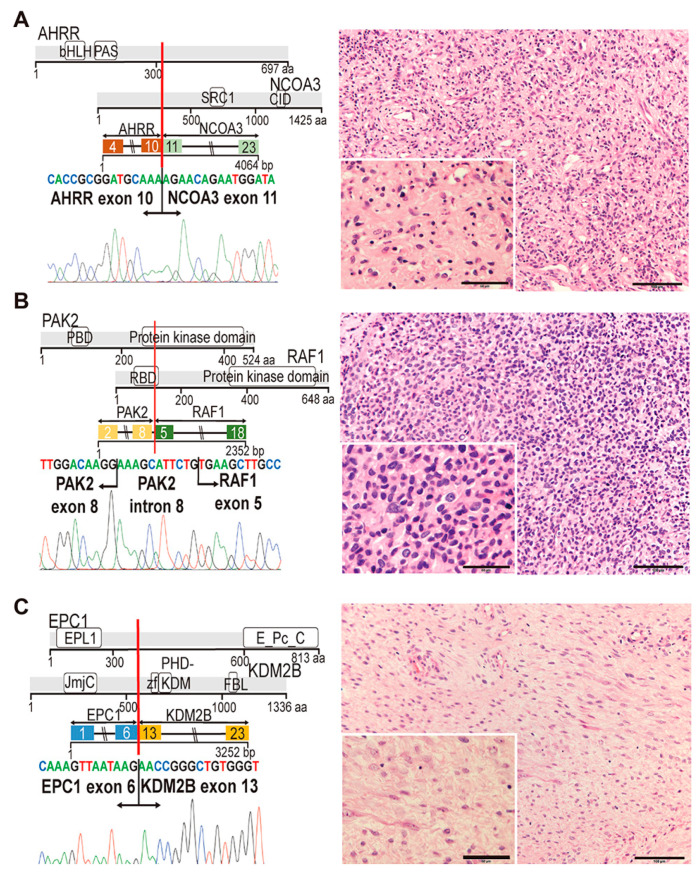
Diagrammatic representation of three novel fusion genes (part 1). (**A**–**C**) Left: proteins involved in individual fusions, fusion transcripts and the junction sequences validated in the RT-PCR experiments. Right: representative morphologic findings. Scale bars: 100 µm (right) and 50 µm (inset). (**A**) *AHRR-NCOA3* in case #43. bHLH: basic helix-loop-helix domain; PAS: PAS domain involved in protein dimerization; SRC1: Steroid receptor coactivator 1 domain; CID: CREB-interacting domain. Spindle cell sarcoma with relatively low cellularity and inflammatory cell infiltration (right). (**B**) *PAK2-RAF1* in case #2. PBD: p21-Rho-binding domain; RBD: Ras-binding domain. Sarcoma shows round cell morphology with nuclear polymorphism (right). (**C**) *EPC1-KDM2B* in case #55. EPL1: enhancer of polycomb-like 1; E_Pc_C: enhancer of polycomb C-terminus; JmjC: jumonji C; ZF: zinc finger; PHD-KDM: PHD finger in lysine-specific demethylase; FBL: F-box-like. Low-grade spindle cell sarcoma with fibromyxoid stroma (right).

**Figure 3 cancers-12-02345-f003:**
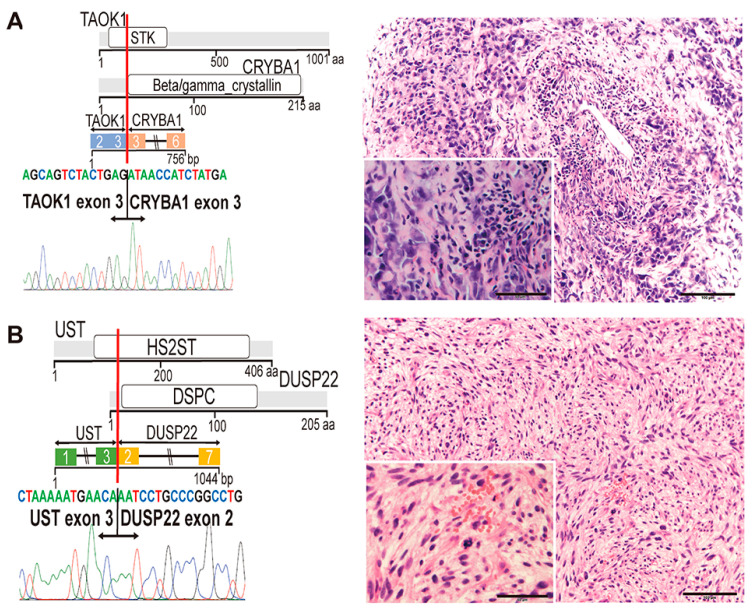
Diagrammatic representation of two novel fusion genes (part 2). (**A**,**B**) Left: proteins involved in individual fusions, fusion transcripts, and the junction sequences validated in the RT-PCR experiments. Right: representative morphologic findings. Scale bars: 100 µm (right) and 50 µm (inset). (**A**) *TAOK1-CRYBA1* in case #14. STK: serine/threonine kinase. Highly aggressive round cell sarcoma with extensive nuclear polymorphism (right). (**B**) *UST-DUSP22* in case #50. HS2ST: heparan sulfate 2-O-sulfotransferase; DSPC: dual-specificity phosphatase catalytic domain. Spindle cell sarcoma with atypical mitosis (right).

**Figure 4 cancers-12-02345-f004:**
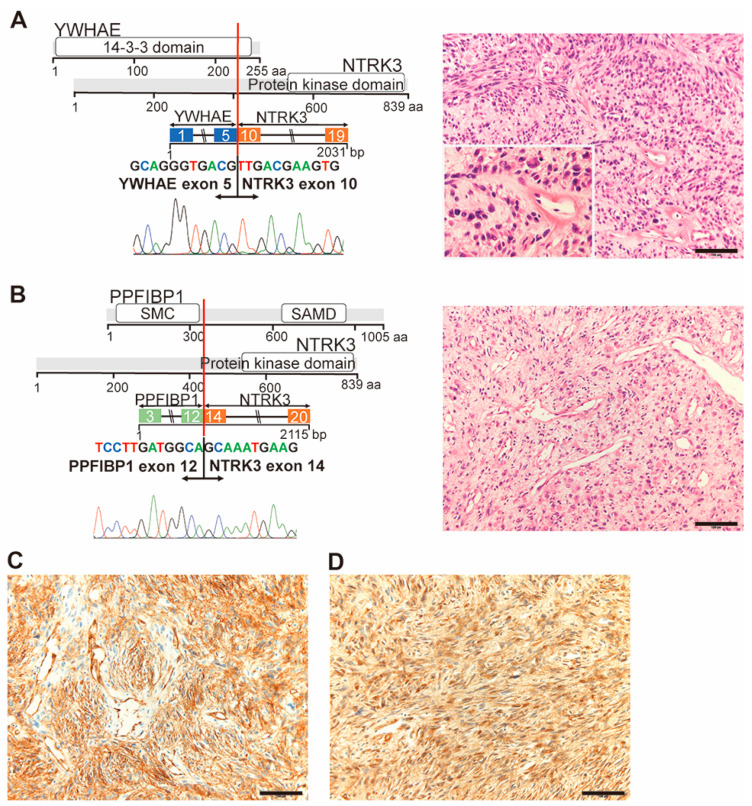
Diagrammatic representation of two novel *NTRK3* fusions. (**A**,**B**) Left: proteins involved in individual fusions, fusion transcripts and the junction sequences validated in the RT-PCR experiments. Right: representative morphologic findings. (**A**) *YWHAE-NTRK3* in case #32. Spindle cell sarcoma with abundant blood vessels (right). The inset shows hyalinization of the blood vessel wall. (**B**) *PPFIBP1-NTRK3* in case #42. SMC: structural maintenance of chromosome; SAMD: sterile alpha motif domain. Short spindle cell sarcoma with abundant blood vessels (right). (**C**,**D**) Immunostaining for case #32 with *YWHAE-NTRK3* using anti-CD34 (**C**) and anti-pan-TRK (**D**) antibodies. Scale bar: 100 µm.

**Figure 5 cancers-12-02345-f005:**
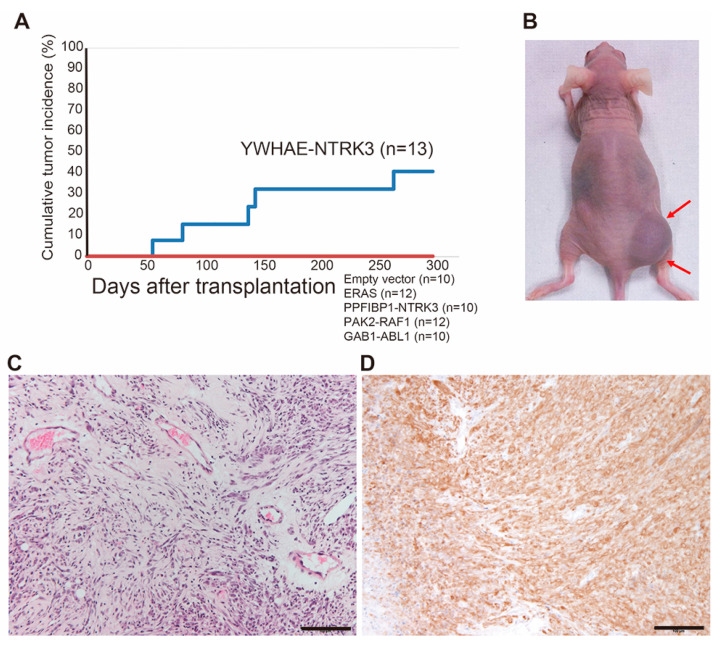
Generation of the mouse model for *YWHAE-NTRK3* sarcoma. (**A**) Cumulative tumor incidence of recipients transplanted with *YWHAE-NTRK3*-expressing mouse embryonic mesenchymal cells (eMCs). Five out of 13 recipients showed tumor development, whereas no tumor was induced by transplantation of *ERAS-, PPFIBP1-NTRK3-, PAK2-RAF1-* or *GAB1-ABL1*-expressing eMCs. (**B**) Subcutaneous tumors (arrows) in the recipient nude mouse. (**C**) Histology of *YWHAE-NTRK3*-expressing sarcoma. (**D**) Immunostaining with an anti-pan-TRK antibody. Scale bar: 100 µm.

**Figure 6 cancers-12-02345-f006:**
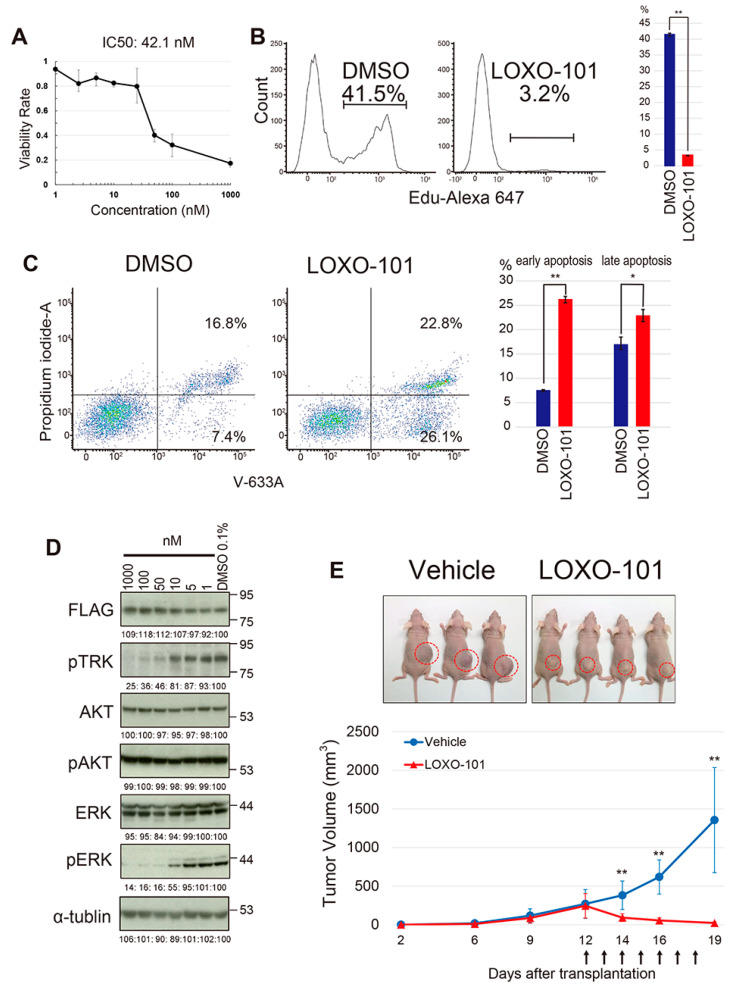
Effects of the NTRK inhibitor LOXO-101 on mouse *YWHAE-NTRK3* sarcoma. (**A**) In vitro growth inhibition of a mouse *YWHAE-NTRK3* sarcoma cell line by LOXO-101. The experiment was performed in triplicate, and average suppression rates with standard error of the mean (SEM) and IC_50_ are indicated. (**B**) Suppression of EdU incorporation in *YWHAE-NTRK3* cells by LOXO-101 treatment. The experiment was performed in triplicate. The representative example is shown in the left panels, and the average EdU incorporation with SEM is shown on the right. (**C**) Detection of apoptosis induced by LOXO-101 treatment for 24 h. Annexin V staining shows a significant increase in both early and late apoptotic cells, as evidenced by flow cytometry (left) and quantified by bar graphs (right). (**D**) Immunoblotting showing decreased phosphorylation of NTRK and ERK by LOXO-101 treatment. Expression of FLAG-tagged YWHAE-NTRK3 is shown in the top panel. (**E**) Growth inhibitory effects of LOXO-101 for mouse *YWHAE-NTRK3* sarcoma in vivo. The sarcoma cells were transplanted subcutaneously into nude mice (top), and tumor volume was measured on the indicated days. The treatment days of LOXO-101 are indicated by arrows (bottom). Seven mice for each group were used. * *p* < 0.01; ** *p* < 0.001.

**Table 1 cancers-12-02345-t001:** Clinicopathological summary of bone and soft tissue sarcoma (BSTS) cases.

Features	Round Cell Sarcomas (*n* = 25)	Spindle Cell Sarcomas (*n* = 30)
Male, sex, *n* (%)	16 (64.0)	15 (50.0)
Age at diagnosis (in years, average ± SD)	50.0 ± 17.0	50.0 ± 19.0
Primary site, *n* (%)		
Extremities	13 (52.0)	20 (66.6)
Trunk	8 (32.0)	10 (33.3)
Head and neck	3 (12.0)	0 (0)
Others	1 (4.0)	0 (0)
Original diagnosis (n)	Round cell sarcoma (14)	Solitary fibrous tumor (12)
	Ewing sarcoma (3)	Synovial sarcoma (6)
	Extraskeletal myxoid chondrosarcoma (3)	Low-grade myxofibrosarcoma (3)
	Clear cell sarcoma (1)	Low-grade fibromyxoid tumor (2)
	Epithelioid hemangioendothelioma (1)	Spindle cell sarcoma (2)
	Malignant epithelioid tumor (1)	Angiofibroma (1)
	Mesenchymal chondrosarcoma (1)	Fibrosarcoma (1)
	Myoepithelial carcinoma (1)	Hemangiopericytoma (1)
		Malignant nodular fasciitis (1)
		Proliferative fasciitis (1)

**Table 2 cancers-12-02345-t002:** Novel fusion genes identified in this study.

Case	Fusion Gene	Chromosome *	Functional Protein
**Round Cell Sarcomas**			
2	*PAK2-RAF1*	inv (3) (p25q29)	yes
6	*YTHDF2-GMEB1*	del (1) (p35p35.3)	no
14	*TAOK1-CRYBA1*	inv (17) (q11.2q11.2)	yes
52	*BTBD9-HMGA2*	t (6;12) (p21;q15)	no
	*BTBD9-PVT1*	t (6;8) (p21;q24)	no
	*PTPRR-KSR2*	inv (12) (q15q24.23)	no
	*TBX5-PTPRR*	del (12) (q24.1q15)	no
**Spindle cell sarcomas**			
21	*DDX26B-IRF4*	t (X;6) (q26.3;p25.3)	no
22	*NCOA2-SERPINA7*	t (X;8) (q22.2;q13.3)	no
32	*YWHAE-NTRK3*	t (15;17) (q25;p13.3)	yes
	*CNNM2-EGR2*	inv (10) (q24.32q21.3)	no
36	*RPSAP52-TGFBR3*	t (1;12) (q14.3;p22.1)	no
38	*MSN-ERAS*	inv (X) (p11.23q12)	yes **
	*FRYL-FIP1L1*	del (4) (p11q12)	no
42	*PPFIBP1-NTRK3*	t (12;15) (p12.1;q25)	yes
	*CRTC1-DOHH*	inv (19) (p13.11p13.3)	no
43	*AHRR-NCOA3*	t (5;20) (q15.3;q12)	yes
50	*UST-DUSP22*	inv (6) (p25.3q25.1)	yes
55	*EPC1-KDM2B*	t (10;12) (p11;q24.31)	yes

* Putative chromosomal abnormalities. ** Production of wild-type ERAS with no MSN component.

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
