# Peer review of "Identification of Novel Fusion Genes in Bone and Soft Tissue Sarcoma and Their Implication in the Generation of a Mouse Model"

_cancers, 2020, doi:10.3390/cancers12092345_

Round 1

Reviewer 1 Report

The authors perform target RNA-seq using a commercial panel (TrueSight RNA Fusion Panel, Illumina) to study a set of 55 bone and soft tissue sarcomas. The authors identify and validate more than 70% of the identified fusion genes and confirm the oncogenic power of one of the novel gene fusion.

  1. The authors affirm that the identified fusions were generated by inter-chromosomal translocations (20/35) or by intra-chromosomal aberrations such as inversions and/or deletions (15/35), but they do not present any data to confirm this affirmation.The authors should consider that some of the fusion genes would not be the result of genomic DNA alterations and discuss other possibilities.
  2. How the authors explain that the gene fusions TAOK1-CRYBA1 and UST_DUSP22 involving genes oriented in the same strand produce, following an chromosome inversion, a in frame gene fusion, or how the PPFIBP1-NTRK3 that are oriented in different strand produce an in frame gene fusion following a translocation.
  3. The authors should describe in more details the analysis performed to identify the fusion genes, for example indicating the threshold of significance used for fusion calling, the QC of the sequencing etc…
  4. How the authors define a novel fusion? Do the authors have used public databases to see in the fusion genes identified have been previously reported; for example checking databases for fusion transcripts
  5. The authors indicate that 7 out of 19 novel fusion genes potentially produce functional fusion proteins. However in table 2 the authors describe 8 (MSN-ERAS )
  6. The authors indicate “Immunoblotting showed suppression of phosphorylation in the NTRK fusion proteins”. However, the molecular weight does not seem to match with the expected molecular weight of the fusion protein (In figure 3A it is indicated that the fusion gene has a length of 2031bp). The molecular weight observed in the western-blot is compatible with the NTRK3 protein western. The authors should clarify these results

Reviewer 2 Report

Using targeted RNA-sequencing technology, the authors have analyzed 55 cases of bone and soft tissue tumors, which were either previously misclassified or unclassified. Besides the non-specific variant histomorphology and immunoprofile, the failure of conventional molecular testing, such as RT-PCR or FISH, in ascertaining the diagnostic classification largely resulted from peculiar exon composition, rare fusion partners or the presence of previously unknown gene fusions. Furthermore, the authors also attempted to generate murine models by explanting embrynonal mesenchymal cells (eMCs) bearing the novel gene fusions that are predicted to constitutively activate tyrosine kinases. Of these, the mouse model bearing the YWHAE-NTRK3 fusion was successfully established, which were histologically proved to recapitulate the spindle morphology in the primary tumor with strong expression of Pan-Trk. In addition, the mouse model was pharmacologically analyzed under LOXO-101 treatment, revealing significant sensitivity to this NTRK inhibitor with apoptosis-inducing and anti-proliferative effects mainly though the blockade of downstream ERK signaling. This paper was well-written and has provided constructive insight into the cancer biology and molecular diagnostics of rare and hard-to-classify bone and soft tissue tumors, while there are a few points in need of further clarification or improvement to upgrade the overall scientific merits.

  1. Regarding Figure 2, I would suggest increase the images of HE sections, preferably from low, medium to high power views, for each representative case in this figure. This will help guide the readers, scientists and pathologists to easily determine whether they are looking at the same type of tumors once they performed RNA-seq and obtained the same gene fusions reported in this study.
  2. In the Table-S2, I would suggest increase a column for revised diagnosis and separately add some representative supplementary histologic images, if available, after the results of functional gene fusions were identified, especially in those harboring previously reported fusions. For instance, case 1 was originally diagnosed as a small round cell sarcoma, while it harbored EWSR1-ATF1. Was this case eventually classified as a clear cell sarcoma, angiomatoid fibrous histiocytoma or other rare entities with this fusion? Case 31 was originally diagnosed as solitary fibrous tumor, while it harbored GAB1-ABL1 previously reported once only in a soft tissue angiofibroma. Case 33 was originally diagnosed as low-grade fibromyxoid sarcoma but its fusion gene turned out to be EWSR1-CREB1, more frequently seen in angiomatoid fibrous histiocytoma and seldom in clear cell sarcoma. Was this case a myxoid variant of angiomatoid fibrous histiocytoma? How about the status of MUC4 staining ?

Round 2

Reviewer 1 Report

The authors have addressed the main points that were raised in the previous review. No further comments.